# GEOMETRY FORCING: MARRYING VIDEO DIFFUSION AND 3D REPRESENTATION FOR CONSISTENT WORLD MODELING

**Haoyu Wu**[1][\*]**, Diankun Wu**[2][\*]**, Tianyu He**[1][†]**, Junliang Guo**[1]**, Yang Ye**[1]**,
Yueqi Duan**[2]**, Jiang Bian**[1]

[1]Microsoft Research    [2]Tsinghua University

## ABSTRACT

Videos inherently represent 2D projections of a dynamic 3D world. However, our analysis suggests that video diffusion models trained solely on raw video data often fail to capture meaningful geometric-aware structure in their learned representations. To bridge the gap between video diffusion models and the underlying 3D nature of the physical world, we propose Geometry Forcing, a simple yet effective method that encourages video diffusion models to internalize 3D representations. Our key insight is to guide the model's intermediate representations toward geometry-aware structure by aligning them with features from a geometric foundation model. To this end, we introduce two complementary alignment objectives: Angular Alignment, which enforces directional consistency via cosine similarity, and Scale Alignment, which preserves scale-related information by regressing geometric features from normalized diffusion representations. We evaluate Geometry Forcing on both camera-view conditioned and action-conditioned video generation tasks. Experimental results demonstrate that our method substantially improves visual quality and 3D consistency over the baseline methods.
Project page: https://GeometryForcing.github.io

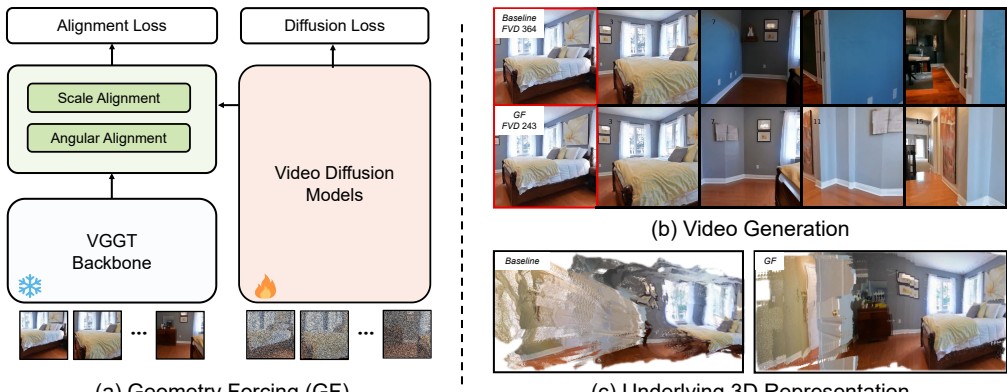

Figure 1: **Geometry Forcing equips video diffusion models with 3D awareness.** **(a)** We propose Geometry Forcing (GF), a simple yet effective paradigm to internalize geometric-aware structure into video diffusion models by aligning with features from a geometric foundation model, *i.e.*, VGGT (Wang et al., 2025b). **(b)** Compared to the baseline method (Song et al., 2025), our method produces more consistent generations both temporally and geometrically. **(c)** Features learned by the baseline model fail to reconstruct meaningful 3D geometry, whereas our method internalizes 3D representation, enabling accurate 3D reconstruction from the intermediate features.

---

[\*]Equal contribution.
[†]Project lead.

# 1 INTRODUCTION

Learning to simulate the physical world and predict future states is a cornerstone of intelligent systems (Ha & Schmidhuber, 2018). Recent advances in generative modeling (Ho et al., 2020; Rombach et al., 2022; Peebles & Xie, 2023; Brown et al., 2020), coupled with the availability of large-scale video datasets, have led to significant progress in generating realistic visual environments conditioned on text descriptions (OpenAI, 2024; Yang et al., 2024; Polyak et al., 2024; Google, 2025) or agent actions (Hu et al., 2023; Guo et al., 2025; Bar et al., 2025). However, these approaches typically aim to model pixel distributions across video frames, overlooking a fundamental principle: *videos are 2D projections of a dynamic 3D world* (Glassner, 1989). By focusing solely on image-space generation, such models often struggle to maintain geometric coherence and long-term consistency, particularly in autoregressive settings where small errors can accumulate over time (Chen et al., 2024a; Cheng et al., 2025; Huang et al., 2025b).

Building on this motivation, a growing line of research has explored the explicit modeling of the dynamic 3D structure of the physical world (Zhu et al., 2024; Aether et al., 2025; Jiang et al., 2025), rather than implicitly learning distributions in 2D pixel space. For example, Zhang et al. (2025a) proposes transforming 3D coordinates into point maps and jointly modeling the RGB and 3D information. While effective to some extent, representing 3D information in a tractable form remains challenging, and reliance on additional annotations limits scalability.

In this work, we aim to bridge the gap between video diffusion models and the underlying dynamic 3D structure of the physical world. We begin with a fundamental question: *Can video diffusion models implicitly learn 3D information through training on raw video data, without explicit 3D supervision*? To investigate this, we analyze a pretrained video diffusion model (Song et al., 2025) by introducing a DPT (Ranftl et al., 2021) head that maps its intermediate features to corresponding depth maps (Wang et al., 2025b). As illustrated in Fig. 1(c), we observe that features learned solely from raw video data fail to yield meaningful geometric representations, highlighting a potential gap in the geometric understanding of video diffusion models trained without additional guidance.

To address this limitation, we propose *Geometry Forcing (GF)*, a simple yet effective approach that encourages video diffusion models to *internalize* 3D representations during training. Inspired by recent advances in semantic REPresentation Alignment (REPA) for image diffusion models (Yu et al., 2024a), we align features of video diffusion models with the *geometric representations* from a pretrained 3D foundation model (Wang et al., 2025b). To align these two representations, our method introduces two complementary alignment objectives: Angular Alignment and Scale Alignment. Angular Alignment enforces directional consistency between the diffusion model's features and geometric representations by maximizing their cosine similarity. Scale Alignment, in contrast, preserves the scale information of the geometric representations by predicting geometric features from normalized diffusion features. The decoupled formulation of Angular and Scale Alignment allows the model to capture both directional and scale-related aspects of geometry, while improving stability during training and expressiveness in the learned representations.

We evaluate the effectiveness of GF on two widely adopted benchmarks: camera-view-conditioned video generation on RealEstate10K (Zhou et al., 2018) and action-conditioned video generation in the Minecraft environment (Baker et al., 2022). Experimental results demonstrate that our method delivers substantial gains in geometric consistency and visual quality over the baseline methods. For example, GF reduces the FVD from 364 to 243 on the RealEstate10K benchmark. Moreover, the ability to reconstruct explicit geometry during inference opens opportunities to integrate structured memory into long-term world modeling.

# 2 RELATED WORK

## 2.1 INTERACTIVE WORLD MODELING

A world simulator seeks to model the underlying dynamics of the physical world by predicting future states conditioned on current observations and conditions. We review prior works through the lenses of interactive video generation, 4D generation, and consistent world modeling.

**Interactive Video Generation.**   Recent advancements in generative models (Ho et al., 2020; Rombach et al., 2022; Peebles & Xie, 2023; Lipman et al., 2023; Bruce et al., 2024; Parker-Holder et al., 2024; Alonso et al., 2024; Valevski et al., 2024) have positioned video generation as a promising approach to world modeling. Beyond text-to-video synthesis (Chen et al., 2023; 2024b; Kong et al., 2024; Wan et al., 2025; Li et al., 2024; Liu et al., 2025a; Ye et al., 2025), interactive video generation (Yu et al., 2025b) that emphasizes responding interactive control signals evolves rapidly. Existing models incorporate different signals like camera controls (He et al., 2024; Yu et al., 2024b; Song et al., 2025) and action controls (Decart et al., 2024; Guo et al., 2025; Feng et al., 2024; Shin et al., 2024). Building on this progress, our work introduces a novel training pipeline that enhances 3D consistency in video generation, enabling more coherent and realistic simulation of spatial scenes.

**Interactive 4D Generation.**   In contrast to data-driven video simulators, 4D-based simulators (Chung et al., 2023; Bahmani et al., 2024b; Wu et al., 2025b; Yu et al., 2025a; Lee et al., 2024) explicitly model dynamic 3D structures  (Kerbl et al., 2023; Mildenhall et al., 2021; Xiang et al., 2025). Building upon 3D content generation (Raj et al., 2023), these methods evolve from dynamic objects (Xu et al., 2024; Bahmani et al., 2024a) to complex dynamic scenes (Niemeyer & Geiger, 2021; Zhu et al., 2024). Recent works integrate video priors to improve the realism and temporal coherence of 4D (Aether et al., 2025; Jiang et al., 2025; Mai et al., 2025; Chen et al., 2025). For example, TesserAct (Zhen et al., 2025) predicts RGB, depth, and surface normals to reconstruct temporally consistent 4D scenes. While our work shares the goal of unifying 3D and video generation, it differs in that it injects 3D geometric priors into the video representation to improve temporal and spatial coherence.

**Consistent World Modeling.**   A key challenge in world modeling lies in maintaining consistency over long video sequences. To address this, prior works have explored different solutions. Frame-level context mechanisms (Chen et al., 2024a; Fuest et al., 2025; Po et al., 2025; Wu et al., 2025c) improve consistency by training with noisy context frames. Meanwhile, other methods leverage 3D information. For example, Xiao et al. (2025) maintains a memory bank indexed by field-of-view overlap to retrieve relevant historical frames. Zhang et al. (2025a) proposes jointly modeling RGB frames and point maps to maintain 3D consistency. In contrast, we propose directly incorporating 3D representations into video diffusion models, thereby enabling more stable geometric consistency.

## 2.2 3D Foundation Models

3D foundation models (3DFMs) (Li et al., 2025; Yang et al., 2025; Smart et al., 2024; Wang* et al., 2025; Wang et al., 2024) have recently shown remarkable progress, applying end-to-end framework with fast and robust inference. These models are capable of predicting different 3D properties, including camera poses (Zhang et al., 2025b), depth maps (Piccinelli et al., 2024), and dense point clouds (Wang et al., 2025b), directly from visual inputs.

Due to their accuracy, efficiency, and robustness, 3DFMs are becoming essential for enabling downstream tasks like spatial reasoning (Wu et al., 2025a; Huang et al., 2025a; Fan et al., 2025), autonomous driving (Fei et al., 2024), SLAM (Liu et al., 2025b; Maggio et al., 2025), and beyond. Inspired by their strong 3D capabilities, we explore incorporating 3D representations into video diffusion models to enhance temporal and spatial consistency for world modeling.

## 3 Preliminaries

Our approach builds upon autoregressive video diffusion models (Chen et al., 2024a; Song et al., 2025; Cheng et al., 2025) and incorporates a 3D foundation model (Wang et al., 2025b) into the training process to guide geometric learning. In this section, we provide a brief overview of both components to establish the foundation for our method.

### 3.1 Autoregressive Video Diffusion Models

**Training.**   We formulate the training pipeline based on Flow Matching (Lipman et al., 2023; Liu et al., 2023) with a Transformer backbone (Vaswani et al., 2017; Bao et al., 2023) to achieve simplicity and scalability. Let $\mathbf{x} = \{x_1, \ldots, x_I\}$ denote a video sequence sampled from the data distri-

bution. We assign an independent timestep for each frame $\mathbf{t} = \{t_1, \ldots, t_I\}$ and corrupt frames via interpolation:

$$x_i^{t_i} = (1 - t_i) \cdot x_i^0 + t_i \cdot \epsilon_i, \quad \text{where} \quad \epsilon_i \sim \mathcal{N}(0, I).$$

The target velocity field is defined as the difference between the noise and the clean input. We train a neural network $v_\theta$ to minimize the Flow Matching loss:

$$\mathcal{L}_{\text{FM}} = \left\| v_\theta(\mathbf{x}^{\mathbf{t}}, \mathbf{t}) - (\boldsymbol{\epsilon} - \mathbf{x}) \right\|^2.$$

**Sampling.** At inference time, the sampling follows a simple probability flow ODE:

$$\mathrm{d}\mathbf{x} = v_\theta(\mathbf{x}^{\mathbf{t}}, \mathbf{t}) \cdot \mathrm{d}\mathbf{t}.$$

In practice, we iteratively apply the standard Euler solver (Euler, 1845) to sample data from noise. For autoregressive generation, we initialize the inputs with a clean context and generate subsequent frames sequentially, conditioning each prediction on the previously generated frames.

## 3.2 VISUAL GEOMETRY GROUNDED TRANSFORMER

Visual Geometry Grounded Transformer (VGGT) (Wang et al., 2025b) is a feed-forward model that directly outputs 3D attributes of a scene, including camera parameters, point maps, and depth maps.

VGGT has a Transformer backbone and multiple prediction heads. The model employs an Alternating-Attention mechanism that interleaves frame-wise self-attention and global self-attention to extract local and global information. For each frame, local and global features are integrated into a unified representation, which is subsequently processed by task-specific heads to produce 3D attributes. We leverage the features from the Transformer backbone of VGGT to extract geometric representation.

## 4 GEOMETRY FORCING

### 4.1 METHOD OVERVIEW

**Motivation.** Recent advances in video diffusion models have enabled the simulation of the world directly from large-scale video datasets. However, these models often overlook a fundamental property of visual data: videos are 2D projections of a dynamic 3D world. To address this, we seek to narrow the gap between video diffusion models and the dynamic 3D structure of the world.

**Observation.** We begin by examining whether video diffusion models are capable of implicitly learning 3D information when trained solely on raw video data, without explicit 3D supervision. To probe the geometric content of their learned representations, we adopt a strategy inspired by linear probing (He et al., 2020): we freeze the parameters of a pretrained video diffusion model (Song et al., 2025) and train a DPT (Ranftl et al., 2021) head to map intermediate features to corresponding depth maps (Wang et al., 2025b). This allows us to assess the extent to which geometric information is encoded in the model's feature space. The results, presented in Fig. 1(c), indicate that features learned solely from raw video data do not produce meaningful geometric representations, suggesting a limited capacity of the model to encode dynamic 3D structure without explicit geometric guidance.

**Challenge.** Bridging the gap between video diffusion models and the dynamic 3D structure of the world presents significant challenges, primarily due to the limited annotated 3D data. A straightforward approach is to jointly model RGB and geometric information within an end-to-end architecture. However, relying heavily on 3D annotations can hinder the scalability and generalization of the models, particularly when applied to large, diverse real-world video datasets.

In this work, inspired by recent advances in REPA (Yu et al., 2024a), we propose *Geometry Forcing (GF)* that aligns the features of video diffusion models with geometric representations, encouraging the model to internalize geometric information. Our approach builds upon video diffusion models described in Sec. 3.1. In Sec. 4.2, we introduce two regularization objectives designed to facilitate representation alignment between the diffusion model and geometric foundation model. The overall training objective, along with additional functional extensions, is summarized in Sec. 4.3.

## 4.2 GEOMETRIC REPRESENTATION ALIGNMENT

To improve the geometric consistency of the learned representations, we introduce two complementary alignment objectives: *Angular Alignment* and *Scale Alignment*. These objectives are designed to align the latent features of the diffusion model with intermediate representations from a pretrained geometric foundation model (Wang et al., 2025b), ensuring both directional consistency and scale preservation of geometric features within the feature space.

**Angular Alignment.** Angular Alignment enforces directional correspondence between the hidden states of the diffusion model, denoted by $h$, and specified target features, denoted by $y$. We select intermediate features from the Transformer backbone of VGGT (Wang et al., 2025b) as $y$, as these features preserve both local and global information within each frame and can be further used to reconstruct various explicit geometric representations. In practice, the target features $y \in \mathbb{R}^{L \times N \times P \times D}$, where $L$ denotes the number of layers, $N$ denotes the number of input images, $P$ denotes the patch count, and $D$ denotes the feature dimension. To achieve Angular Alignment, we first use a lightweight projector $f_\phi$ to map the diffusion latents $h \in \mathbb{R}^{N \times P' \times D'}$ to $y$'s shape. The Angular Alignment loss is then defined as:

$$\mathcal{L}_{\text{Angular}} = -\frac{1}{LNP} \sum_{\ell=1}^{L} \sum_{n=1}^{N} \sum_{p=1}^{P} \cos\left(y_{\ell,n,p},\ f_\phi(h_{n,p})\right),$$

where $\cos(\cdot, \cdot)$ denotes cosine similarity. This loss aligns hidden states independently at both the frame and patch levels. Since the VGGT backbone already incorporates cross-frame attention, we do not explicitly enforce global alignment across frames in the loss.

**Scale Alignment.** While Angular Alignment ensures directional consistency, it disregards feature scale, which can also encode geometric information. Although direct mean squared error (MSE) loss could supervise magnitudes, it often leads to optimization instability and model collapse due to inherent scale differences across models. To address this issue, we introduce Scale Alignment, which preserves scale information through predicting the scale of target features given normalized diffusion hidden states. Specifically, we first normalize $f_\phi(h)$ to unit length. Then we use another lightweight prediction head $g_\varphi$ to predict the full target features from normalized inputs:

$$\hat{h}_{\ell,n,p} = \frac{f_\phi(h_{n,p})}{\|f_\phi(h_{n,p})\|_2}, \quad \tilde{y}_{\ell,n,p} = g_\varphi(\hat{h}_{\ell,n,p}).$$

The Scale Alignment loss is defined as:

$$\mathcal{L}_{\text{Scale}} = \frac{1}{LNP} \sum_{\ell=1}^{L} \sum_{n=1}^{N} \sum_{p=1}^{P} \|\tilde{y}_{\ell,n,p} - y_{\ell,n,p}\|_2^2.$$

This decomposition stabilizes training while capturing both directional and scale attributes of geometric representations.

## 4.3 3D-AWARE AUTOREGRESSIVE VIDEO DIFFUSION MODELS

Building on the autoregressive video diffusion framework and the proposed alignment objectives, we now present the overall training objective:

$$\mathcal{L} = \mathcal{L}_{\text{FM}} + \lambda_{\text{Angular}} \cdot \mathcal{L}_{\text{Angular}} + \lambda_{\text{Scale}} \cdot \mathcal{L}_{\text{Scale}}.$$

Given the intermediate features of our model are well-aligned with geometric representations, an appealing consequence is the model's ability to predict explicit 3D geometry during inference. This enables unified generation of both video and 4D, effectively bridging the gap between videos and the underlying dynamic 3D structure of the physical world, as illustrated in Fig. 1. Moreover, the ability to reconstruct explicit geometry during inference provides a structured and interpretable form of memory that can be further used to support long-term world modeling. We leave the exploration of such geometry-based memory mechanisms as a promising direction for future work.

Table 1: Quantitative comparison on the RealEstate10K dataset for both short-term (16-Frame) and long-term (256-Frame) video generation. Geometry Forcing substantially improves over the baseline. **bold** values denote the best, and Underlined values indicate the second best. **\*** indicates the method is conditioned on the first frame only.

| Method | Frames | FVD↓ | LPIPS↓ | SSIM↑ | PSNR↑ | RPE↓ | RVE↓ |
|---|---|---|---|---|---|---|---|
| DFoT (Song et al., 2025) | 16 | 252 | 0.40 | 0.50 | 14.40 | – | – |
| REPA (Yu et al., 2024a) | 16 | 221 | 0.37 | 0.54 | **15.20** | – | – |
| VideoREPA (Zhang et al., 2025c) | 16 | 210 | 0.37 | 0.54 | **15.20** | – | – |
| Geometry Forcing (ours) | 16 | 193 | **0.32** | **0.58** | 14.70 | – | – |
| Geometry Forcing (ours) + REPA | 16 | **179** | 0.34 | 0.54 | 15.00 | – | – |
| Cosmos* (Agarwal et al., 2025) | 256 | 934 | 0.68 | 0.20 | 10.25 | – | – |
| DFoT (Song et al., 2025) | 256 | 364 | 0.55 | 0.36 | 11.40 | 0.3575 | 297 |
| REPA (Yu et al., 2024a) | 256 | 297 | 0.54 | 0.36 | 11.51 | 0.3337 | 315 |
| VideoREPA (Zhang et al., 2025c) | 256 | 455 | 0.56 | 0.35 | 11.50 | 0.3823 | **190** |
| Geometry Forcing (ours) | 256 | 243 | **0.51** | **0.38** | 11.87 | 0.3337 | 272 |
| Geometry Forcing (ours) + REPA | 256 | **237** | **0.51** | 0.37 | **12.10** | **0.3264** | 236 |

**Discussion.** Teacher Forcing (Williams & Zipser, 1989) is a widely adopted training paradigm for autoregressive models (Radford et al., 2019; Brown et al., 2020; Kondratyuk et al., 2024). To combine the autoregressive nature with diffusion models, Diffusion Forcing (Chen et al., 2024a) proposes training video diffusion models with independent noise levels for each frame. More recently, Self Forcing (Huang et al., 2025b) has been proposed to address exposure bias in autoregressive video diffusion models. Orthogonal to these methods, Geometry Forcing focuses on improving the structure of the learned representations by aligning the intermediate representation of video diffusion models with geometry-aware signals from a 3D foundation model. Our approach provides structural supervision at the representation level, encouraging the model to internalize 3D consistency throughout training.

# 5 EXPERIMENTS

In this section, we evaluate Geometry Forcing (GF) on camera-view-conditioned video generation on the RealEstate10K (Zhou et al., 2018) dataset and action-conditioned video generation on the Minecraft environment (Baker et al., 2022). We provide additional illustrations and visualizations in the Appendix.

**Implementation Details.** For camera view-conditioned video generation, we apply GF on Diffusion Forcing Transformer (Song et al., 2025), training on 16-frame 256×256 videos for 2,500 steps with a learning rate of $8 \times 10^{-6}$ and batch size 8. Inference is conditioned on the first frame and per-frame camera poses. For action-conditioned video generation, we apply GF to Next-Frame Diffusion (Cheng et al., 2025), training on 32-frame 384×224 videos for 2,000 steps with a learning rate of $6 \times 10^{-5}$ and batch size 32. We set $\lambda_{\text{Angular}} = 0.5$ and $\lambda_{\text{Scale}} = 0.05$ to balance each loss component. All experiments are conducted on 8 NVIDIA A100 GPUs.

**Evaluation Metrics.** We evaluate visual quality using FVD (Fréchet Video Distance) (Unterthiner et al., 2018), PSNR (Peak Signal-to-Noise Ratio), SSIM (Structural Similarity Index) (Wang et al., 2004), and LPIPS (Learned Perceptual Image Patch Similarity) (Zhang et al., 2018).

To further evaluate geometric consistency, we introduce Reprojection Error (RPE) (Duan et al., 2025) and Revisit Error (RVE) (Xiao et al., 2025). Reprojection Error (RPE) quantitatively measures multi-view geometric consistency by calculating the average reprojection discrepancy between projected and observed pixel locations across frames. Revisit Error (RVE) assesses long-range temporal consistency by examining discrepancies between initial and revisited frames under complete camera rotation. We provide more details of these metrics in the Appendix (Sec. C.4).

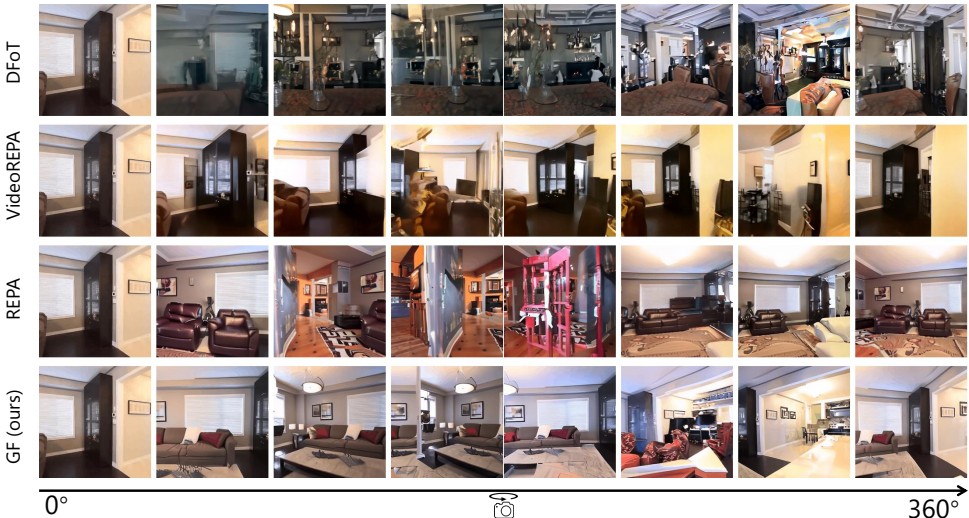

Figure 2: **Qualitative comparison of camera view-conditioned video generation under full-circle rotation.** Videos are generated from a single frame, and per-frame camera poses simulate a full 360° rotation. Our method (GF) is compared with DFoT (Song et al., 2025), VideoREPA (Zhang et al., 2025c), and REPA (Yu et al., 2024a). The results demonstrate that the baseline methods fail to maintain temporal consistency, while our proposed GF consistently revisits the starting viewpoint.

## 5.1 MAIN RESULTS

This section presents the main experimental results, comparing our method against state-of-the-art approaches across different tasks. The evaluation results demonstrate the effectiveness and generalization ability of our method in both short- and long-term video generation.

**Camera view-conditioned Video Generation.** We conduct a comprehensive evaluation of GF on the RealEstate10K (Zhou et al., 2018) dataset, comparing against state-of-the-art baselines. We report results for both short-term (16-Frame) and long-term (256-Frame) video generation in Tab. 1. As shown in Tab. 1, our method consistently outperforms all baselines across multiple evaluation metrics, including FVD, LPIPS, SSIM, and PSNR, in both the short-term and long-term generation settings. These results highlight the effectiveness of GF in enhancing visual fidelity, temporal stability, and 3D spatial consistency, thereby enabling more realistic and coherent world modeling.

**Action-conditioned Video Generation.** To demonstrate the generality of our method, we apply GF to Next-Frame Diffusion (Cheng et al., 2025) model. As shown in Tab. 5, the model achieves a lower FVD score, indicating that GF can be seamlessly integrated into video diffusion models and yields measurable gains. Note that there exists a large data distribution gap between the real world and Minecraft. These results demonstrate that GF generalizes well on out-of-domain distributions.

## 5.2 QUALITATIVE RESULTS

Fig. 2 presents qualitative comparisons on the RealEstate10K dataset. Each video is generated from a single input frame along with per-frame camera poses simulating a full 360° rotation. We compare GF against three strong baselines: DFoT (Song et al., 2025), REPA (Yu et al., 2024a), and VideoREPA (Zhang et al., 2025c). As shown in Fig. 2, our method reconstructs the initial frame upon completion of the camera rotation, while producing reasonable and realistic intermediate views. In contrast, the baseline methods fail to maintain temporal coherence and scene consistency, resulting in implausible intermediate frames and an inability to revisit the starting viewpoint. These results highlight the superior long-term 3D consistency and scene understanding of our approach.

## 5.3 ABLATION STUDIES

We provide a series of ablation studies to validate the design of GF.

Table 2: **Ablation study on target representation**. We compare the effect of aligning the diffusion model with different target representations: DINOv2 (semantic), VGGT (geometric), and their combination. The joint use of both representation achieves the best FVD.

| Target Representation | FVD-256 |
|---|---|
| Baseline | 364 |
| DINOv2 Only | 297 |
| VGGT Only | 243 |
| VGGT + DINOv2 | **237** |

Table 3: **Ablation study on alignment loss.** Angular and Scale Alignment losses are evaluated for long-term video generation, with MSE as a naive baseline of aligning both angular and scale information. The combination of Angular and Scale Alignment yields the best results.

| Alignment Loss | FVD-256 |
|---|---|
| Baseline | 364.0 |
| Angular | 253.0 |
| Angular + Scale | **243.0** |
| MSE | 1648.0 |

Table 4: **Ablation study on explicit and implicit geometry information.** We compare the explicit geometry condition with internal alignment (ours).

| Method | FVD-256$\downarrow$ |
|---|---|
| Baseline | 364 |
| Explicit geometry | 280 |
| Geometry Forcing (ours) | **243** |

Table 5: **Evaluation on action-conditioned video generation in Minecraft.** FVD results of NFD before and after applying Geometry Forcing (GF) on 16-Frame generation show clear improvement.

| Method | FVD-16$\downarrow$ |
|---|---|
| NFD | 216 |
| NFD + GF | **205** |

**Which Representation Should be Aligned?** To validate the effectiveness of geometric representation, we compare two target representations in GF: VGGT (Wang et al., 2025b), trained on 3D datasets with strong geometric priors, and DINOv2 (Oquab et al., 2023), trained on 2D images focusing on semantic features. As shown in Tab. 2, aligning with VGGT consistently outperforms DINOv2 on both long-term and short-term generation tasks, highlighting the advantage of geometric alignment over semantic supervision.

To further explore their complementarity, we combine VGGT and DINOv2 features as joint supervision targets. Results in Tab. 2 show that integrating geometric and semantic signals leads to additional gains, suggesting that the two types of representations are orthogonal and can enhance each other when used together. However, as we mainly focus on bridging the gap between the video diffusion model and the dynamic 3D structure of the real world, we use only VGGT features in subsequent experiments.

**Alignment Loss.** GF consists of two alignment objectives: Angular Alignment and Scale Alignment. To validate their effectiveness, we compare three alignment loss types: (1) Angular Alignment alone (Sec. 4.2), (2) Angular Alignment with Scale Alignment (Sec. 4.2), and (3) MSE loss between VGGT and diffusion features. As shown in Tab. 3, the combination of Angular Alignment and Scale Alignment achieves the best performance, indicating the benefit of aligning both angular and scale-related information. Although direct mean squared error (MSE) also supervises magnitudes, changes in the diffusion model's feature scale may cause collapse in the following layers. These results highlight that neither Angular Alignment nor Scale Alignment alone is sufficient.

**Explicitly or Implicitly Integrate Geometry Information into Video Diffusion Models?** To assess the benefit of internalizing geometric representations, we compare two ways of incorporating geometry into the video diffusion model: internal alignment via GF and external guidance via a ControlNet (Zhang et al., 2023). For external guidance, we reconstruct the 3D scene, render it into 2D images, and inject the rendered images as geometric conditions. As shown in Tab. 4, using the same VGGT features, GF outperforms rendered-image conditioning. The result shows that while explicit geometric cues are helpful, internal alignment through GF provides consistently stronger supervision. By aligning internal features with geometric representations, GF enables a deeper geometric understanding and yields better performance in perceptual quality and structural consistency. Full evaluation results are listed in Tab. 8.

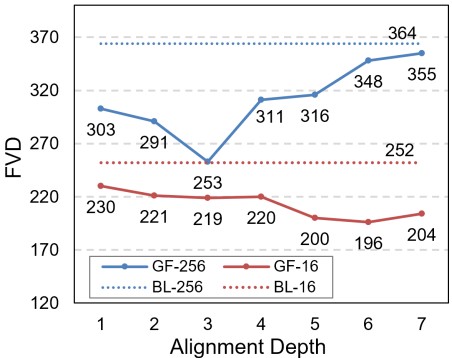

Figure 3: **Ablation study on alignment depth.** We present FVD-256 and FVD-16 results for different alignment layers of the diffusion model, which suggest that mid-level features are most effective to improve video quality.

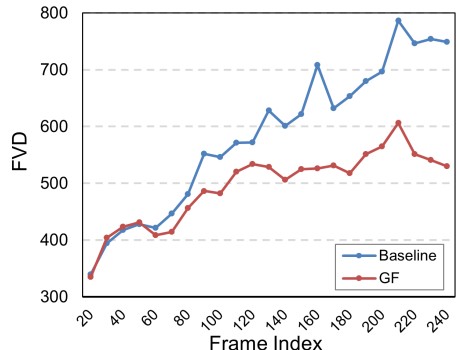

Figure 4: **Exposure bias analysis.** This figure shows the trend of FVD scores during long-term video generation. Compared to the baseline, GF results in significantly lower FVD after 100 frames.

Table 6: **User study.** Average scores on Camera Following, Object Consistency, and Scene Continuity. Each user has to rate each dimension on a scale of 1 to 5. Higher values indicate better quality.

| Method | Camera Following | Object Consistency | Scene Continuity |
|---|---|---|---|
| DFoT | 3.56 | 2.73 | 2.74 |
| REPA | 3.82 | 3.55 | 3.66 |
| VideoREPA | 3.31 | 3.05 | 2.82 |
| Geometry Forcing | **4.40** | **4.44** | **4.52** |

**Which Layer Should be Aligned?** As shown in Fig. 3, we also explore applying alignment at different layers of the video diffusion model (Song et al., 2025), which uses a 7-layer U-ViT (Bao et al., 2023) backbone (3 downsampling layers, 1 bottleneck layer, 3 upsampling layers). Aligning at layer 3 yields the best FVD-256 score while preserving FVD-16 performance.

**Mitigating Exposure Bias in Autoregressive Video Diffusion Model via Geometry Forcing.** Exposure bias is a long-standing challenge in autoregressive video generation (Chen et al., 2024a; Song et al., 2025; Sun et al., 2025; Cheng et al., 2025; Huang et al., 2025b). While previous methods attempted to address it through memory mechanisms or context guidance, GF offers an orthogonal solution. As shown in Fig. 4, GF mitigates long-term drift and reduces the accumulation of error during generation significantly by aligning 3D geometric representation. These results validate integrating 3D representation enables more reliable and coherent long-term video synthesis.

## 5.4 USER STUDY

While Reprojection Error (RPE) and Revisit Error (RVE) provide useful signals for measuring 3D consistency, they only capture specific geometric aspects and may miss perceptual artifacts or unrealistic dynamics that humans can easily notice. We therefore conduct a user study focusing on three aspects of 3D consistency. 1) **Camera Following**: Whether the camera in the video moves smoothly and accurately follows the given pose trajectory. 2) **Object Consistency**: Whether objects remain consistent in shape, appearance, and position across frames. 3) **Scene Continuity**: Whether the generated parts of the scene beyond the context frames remain coherent and reasonable.

We compare GF with DFoT (Song et al., 2025), REPA (Yu et al., 2024a), and VideoREPA (Zhang et al., 2025c). As shown in Tab. 6, GF consistently outperforms all baselines across the three aspects of 3D consistency, demonstrating its effectiveness in producing geometrically coherent videos.

## 6 CONCLUSION

This paper introduces Geometry Forcing (GF), a simple yet effective framework that enhances the geometric consistency of autoregressive video diffusion models by aligning their internal representations with geometry-aware features. Motivated by the observation that video diffusion models trained on raw pixel data often fail to capture meaningful 3D structure, our method introduces two alignment objectives, Angular Alignment and Scale Alignment, to guide latent features toward 3D-aware representations from a geometric foundation model. Empirical results on both camera-conditioned and action-conditioned video generation benchmarks demonstrate that GF significantly improves visual quality and 3D consistency, yielding lower FVD scores and more stable scene dynamics.

**Limitations.** The primary limitation of this work lies in its scale. While GF consistently improves geometric consistency and visual quality, its full potential remains unexplored under large-scale training. In particular, we have not yet investigated its effectiveness when applied to larger models and more extensive video datasets, which may further amplify its benefits.

**Future Work.** Future directions include scaling GF on larger datasets to build 3D-consistent world simulators, and applications for long video generation by treating 3D representation as memory.

## ACKNOWLEDGMENT

We would like to acknowledge Kiwhan Song and Boyuan Chen for their valuable advice and assistance in reproducing the Diffusion Forcing Transformer results.

REPRODUCIBILITY

We provide comprehensive implementation details, including model architectures, training configurations, and data preprocessing procedures, in Appendix C to ensure reproducibility.

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

## A   DECLARATION OF LLM USAGE

We used large language models (LLMs) to aid or polish writing. Details are described in the paper. The use is limited to language editing (grammar, spelling, and word choice), code formatting (e.g., adding comments to the code). All scientific ideas, analysis, and conclusions were conceived, validated, and interpreted independently by the authors. We gratefully acknowledge the assistance of large language models in our work.

## B   LIMITATIONS

Our method's reliance on VGGT (trained mainly on static scenes) constrains performance in dynamic environments. Generalization to significant motion scenarios requires further research.

## C  IMPLEMENTATION DETAILS

### C.1  DATASET

**RealEstate10K (Zhou et al., 2018).**  This dataset contains camera poses for 10 million video frames, suitable for evaluating 3D consistency and camera navigation in generated videos. We use a resolution of $256 \times 256$ pixels.

**Minecraft (Baker et al., 2022).**  This game dataset includes action annotations, enabling evaluation of video generation in dynamic environments with camera motion.

**Alignment Projection.**  To maximize geometric information retention, we aggregate features from all transformer blocks of the VGGT backbone as alignment targets. For computational efficiency, we apply bilinear interpolation to reduce the spatial dimensions from the original resolution to a manageable $512 \times 512$ size.

The alignment is performed using a Conv3D-based projector that operates on the latent dimensions. To accommodate multi-layer and multi-target alignment scenarios, we initialize independent projectors for each feature layer and target representation. This design ensures effective dimensional compatibility between the U-ViT feature space and the target geometric representations while maintaining computational efficiency.

### C.2  TRAINING

**Model Architecture.**  We adopt a U-ViT (Bao et al., 2023) backbone for video generation, with geometric feature alignment integrated at the third transformer block.

**Training Data.**  The model is trained on 10,000 video clips sampled from the RealEstate10K training dataset, each comprising 16 consecutive frames.

**Training Protocol.**  Training proceeds for 2 epochs using a learning rate of $8 \times 10^{-6}$ and a global batch size of 40. The geometric alignment loss is combined with the standard diffusion training objective.

### C.3  INFERENCE

A key advantage of Geometry Forcing is its inference-time efficiency, which introduces no computational overhead during sampling. We demonstrate results using a DDIM sampler with 50 steps, though the approach is compatible with any standard diffusion sampling algorithm.

### C.4  METRICS

In this section, we present the detailed implementations of the Reprojection Error (RPE) and the Revisit Error (RVE).

**Reprojection Error.**  Reprojection error (RPE) is a widely used metric in visual SLAM to evaluate multi-view geometric consistency. Following Duan et al. (2025), we utilize DROID-SLAM (Teed & Deng, 2021) to reconstruct the scene. Specifically, DROID-SLAM first extracts corresponding features across frames and then refines camera poses ($G_t$) and per-pixel depth estimates ($d_t$) through its differentiable Dense Bundle Adjustment (DBA) optimization, enforcing optical flow constraints and achieving robust structure-from-motion. The reprojection error is then computed by measuring the average Euclidean distance between the projected and observed pixel locations of co-visible 3D points across multiple frames. Formally, RPE is defined as:

$$RE = \frac{1}{|\mathcal{V}|} \sum_{(i,j) \in \mathcal{V}} \left\| \mathbf{p}_{ij}^* - \Pi(\mathbf{P}_{ij}) \right\|_2 , \tag{1}$$

where $\mathcal{V}$ denotes the set of valid feature correspondences, $\mathbf{p}{ij}$ is the observed pixel location in generated video frames, $\mathbf{P}{ij}$ represents the corresponding reconstructed 3D point derived from refined

Table 7: **Ablation study on teacher model.** Our method (Geometry Forcing) is compatitable with different teacher models including VGGT and Pi3. **Bold** values denote the best, and Underlined values indicate the second best. * indicates the method is conditioned on the first frame only.

| Method | Frames | FVD↓ | LPIPS↓ | SSIM↑ | PSNR↑ | RPE↓ | RVE↓ |
|---|---|---|---|---|---|---|---|
| DFoT (Song et al., 2025) | 256 | 364 | 0.55 | 0.36 | 11.40 | 0.3575 | 297 |
| Pi3 Wang et al. (2025) | 256 | 309 | 0.53 | **0.38** | 11.53 | **0.3171** | 303 |
| Geometry Forcing (VGGT) | 256 | **243** | **0.51** | **0.38** | **11.87** | 0.3337 | **272** |

depths and camera poses, and Π denotes the camera projection function. Lower RPE values indicate better 3D alignment, reduced spatial artifacts, and enhanced spatio-temporal stability, thereby effectively reflecting the overall geometric coherence and consistency of the generated videos.

**Revisit Error.** Revisit Error evaluates long-range temporal consistency under full camera rotation, inspired by the setup proposed in WorldMem (Xiao et al., 2025). For each of 100 randomly sampled RealEstate10K video clips, we extract the first frame and initial camera pose. A camera trajectory of 256 frames is then constructed by rotating the initial camera pose around the Y-axis. We assess revisit consistency by comparing the first and final frame using reconstruction FID (rFID) (Heusel et al., 2017). Larger discrepancies indicate greater geometric or appearance drift, suggesting weaker long-term 3D consistency.

## C.5 3D Reconstruction from Diffusion Features

In this section, we provide a detailed overview of the 3D reconstruction process illustrated in Fig. 1(c).

**Reconstruction using Geometry Forcing Features.** We extract features from the Geometry Forcing (GF) model and pass them through the depth prediction head of VGGT to obtain the predicted depth map.

**Reconstruction using Diffusion Features.** Motivated by our linear probing experiments, we investigate the 3D reconstruction capability of intermediate features extracted from DFoT (Song et al., 2025). Specifically, we freeze the pretrained DFoT backbone and train a DPT head (Ranftl et al., 2021) to regress depth maps from its intermediate representations. The target depth maps are provided by the VGGT model (Wang et al., 2025b) and serve as ground-truth supervision. The DPT head adopts the same architecture as the depth prediction module used in VGGT but is trained from scratch. We optimize the DPT head for 2500 steps using a learning rate of $1 \times 10^{-4}$ and a batch size of 4.

## D Supplementary Experiments

### D.1 Ablation on Teacher Model

Geometry Forcing does not depend on a specific 3D foundation model but still requires the 3D foundation to be feed-forward and to support multi-image inputs, as required by online training. We conduct Geometry Forcing algorithm on $Pi^3$ model and also achieves significant improvement on video generation as shown in Tab. 7.

### D.2 Explicit Geometry Control

We provide a full evaluation comparison between explicit control and our Geometry Forcing in Tab. 8.

Table 8: **Ablation study on explicit and implicit geometry information.** Our method (Geometry Forcing) achieves the best performance across all metrics on the RealEstate10K dataset for long-term (256-Frame) video generation. **bold** values denote the best, and Underlined values indicate the second best. **\*** indicates the method is conditioned on the first frame only.

| Method | Frames | FVD↓ | LPIPS↓ | SSIM↑ | PSNR↑ | RPE↓ | RVE↓ |
|---|---|---|---|---|---|---|---|
| DFoT (Song et al., 2025) | 256 | 364 | 0.55 | 0.36 | 11.40 | 0.3575 | 297 |
| Explicit geometry | 256 | 280 | 0.52 | 0.37 | **11.99** | 0.3792 | 297 |
| Geometry Forcing (ours) | 256 | **243** | **0.51** | **0.38** | 11.87 | **0.3337** | **272** |

Table 9: **Ablation study on GF alignment context length.** Geometry Forcing-n indicates n frames is used to extract VGGT feature during training. The results are evaluated on the RealEstate10K dataset for long-term (256-Frame) video generation. **bold** values denote the best, and Underlined values indicate the second best. **\*** indicates the method is conditioned on the first frame only.

| Method | Frames | FVD↓ | LPIPS↓ | SSIM↑ | PSNR↑ | RPE↓ | RVE↓ |
|---|---|---|---|---|---|---|---|
| DFoT (Song et al., 2025) | 256 | 364 | 0.55 | 0.36 | 11.40 | 0.3575 | 297 |
| Geometry Forcing-4 | 256 | 261 | 0.51 | **0.38** | **12.21** | 0.3451 | 297 |
| Geometry Forcing-8 | 256 | 257 | **0.50** | **0.38** | 12.17 | **0.3062** | 284 |
| Geometry Forcing-16 (default) | 256 | **243** | 0.51 | **0.38** | 11.87 | 0.3337 | **272** |

## D.3 Alignment Context Length

Geometry Forcing feeds 16 frames into the VGGT model to extract a latent representation, then aligns the first 16 frames during training. We present ablation results for different alignment context lengths in Tab. 9. The results indicate that when the alignment context length is longer, the 3D information is more complete, thus leading to better results.

## D.4 Multiple Layer Alignment

Due to the large number of possible layer combinations for the layers selected for alignment, we present results for aligning the last three layers of our diffusion model in Tab. 10. However, an increasing number of layers to align doesn't lead to better performance.

## D.5 Text-to-video generation

We extend our Geometry Forcing method to general text-to-video generation tasks. Our model is trained on 2K videos from the Wang et al. (2025a), which provide detailed scene and camera descriptions. Experimental results demonstrate that our approach achieves improvements across multiple evaluation dimensions, including visual aesthetics, motion smoothness, and motion quality, as detailed in Table 11. These results indicate that Geometry Forcing can extend effectively to dynamic text-to-video training, even though VGGT itself is trained on static scenes.

## E  Discussion

### E.1  Computational Efficiency

We perform detailed profiling of our method on an NVIDIA A800 GPU and report the execution time and floating-point operations (FLOPs) for different components of our model during training in Table 12. The VGGT Feature Alignment contributes an additional 52.5% in execution time and 60.4% in total FLOPs. Although this alignment process increases the per-step computation compared to the base diffusion model, it significantly accelerates convergence, thereby reducing the overall training duration. For fine-tuning, our method requires only a few thousand steps and completes within hours, yielding substantial efficiency gains over full pre-training. Additionally, during inference, our method incurs no additional computational cost compared to other methods that use explicit or implicit memory.

Table 10: **Alignment on Multiple Layers.** Comparison of aligning VGGT features at the middle layer vs. the last three layers of the diffusion model using Geometry Forcing.

| Method | Frames | FVD↓ | LPIPS↓ | SSIM↑ | PSNR↑ | RPE↓ | RVE↓ |
|---|---|---|---|---|---|---|---|
| DFoT (Song et al., 2025) | 256 | 364 | 0.55 | 0.36 | 11.40 | 0.3575 | 297 |
| Geometry Forcing (Last 3 Layers) | 256 | 280 | 0.52 | 0.37 | **11.99** | 0.3792 | 297 |
| Geometry Forcing (Mid) | 256 | **243** | **0.51** | **0.38** | 11.87 | **0.3337** | **272** |

Table 11: **Evaluation on text-conditioned video generation.** Evaluation on text-conditioned video generation. We report aesthetic quality, imaging quality, and motion smoothness for Wan2.1-1.3B before and after applying Geometry Forcing.

| Method | Aesthetic Quality↑ | Imaging Quality↑ | Motion Smoothness↑ |
|---|---|---|---|
| Wan2.1 | 0.58 | 0.56 | 0.98 |
| Wan2.1 + GF | **0.59** | **0.59** | **0.99** |

We also provide a feature extraction time of the VGGT model in Fig. 5. The result shows that the extraction time increases from 0.1s to 0.8s when the input increases from 1 to 12.

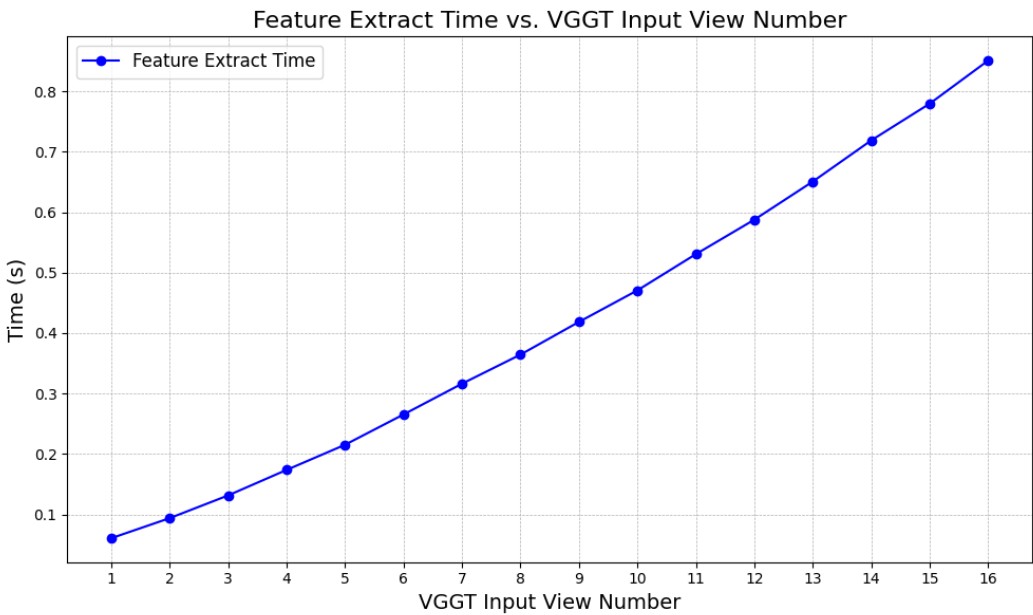

Figure 5: **VGGT Feature Extraction Time.** The feature extraction time of the VGGT model increases with the number of input views.

## E.2 ANALYSIS OF GEOMETRIC AND SEMANTIC REPRESENTATIONS

We analyze the roles of geometric and semantic representation alignment in video generation. First, these representations exhibit considerable overlap rather than orthogonality. Semantic representations like DINOv2 (Oquab et al., 2023) demonstrate zero-shot depth estimation capabilities (see Section 7.5 and Figure 7 in the original paper), indicating inherent geometric understanding. Conversely, geometric representations such as VGGT utilize DINOv2 features as inputs, thereby encoding semantic information.

Second, experimental results in Table 1 and Table 2 show that VGGT alignment primarily enhances 3D consistency, while DINOv2 alignment improves visual quality. The combination of both representations achieves superior performance compared to either individual approach.

Table 12: **Training Stage Profiling.** We report the execution time and floating-point operations (FLOPs) for different components during a training step.

| Pipeline Stage | Time | | FLOPs | |
|---|---|---|---|---|
| | Value (s) | Percentage (%) | Value (T) | Percentage (%) |
| *Forward (Frozen)* | | | | |
| VGGT Encoding | 0.853 | 53.4% | 93.3 | 60.4% |
| *Forward (Learnable)* | | | | |
| Projector | 0.017 | 1.1% | 0.1 | 0.1% |
| Diffusion Backbone | 0.220 | 13.8 % | 17.7 | 11.5% |
| *Backward (Learnable)* | | | | |
| Projector + Diffusion Backbone | 0.506 | 31.7% | 43.5 | 28.1% |
| **Total per Step** | **1.597** | **100.0%** | **154.6** | **100.0%** |

Finally, the distinct contributions of each representation can be characterized as follows: semantic alignment enhances object realism and visual detail, whereas geometric alignment ensures structural consistency and shape coherence across generated video sequences.

### E.3  3D CONSISTENCY AND EXPOSURE BIAS MITIGATION

As shown in Figure 4, the FVD metric increases at a slower rate when Geometry Forcing is employed, indicating effective mitigation of exposure bias in long-term video generation. The underlying mechanism can be understood through the inherent stability of 3D scenes: while the number of generated frames increases, the underlying scene geometry remains the same. Geometry Forcing enables the model to internalize this geometric consistency, thereby reducing error accumulation when regenerating frames from previously encountered viewpoints.

### E.4  FAILURE CASE ANALYSIS

Although our method significantly improves visual quality and geometric consistency in video generation, they still struggle in certain complex scenarios. As shown in Fig. 6, the transparent, reflective glass table intermittently disappears and reappears across frames, indicating that the model still has difficulty handling reflective materials.

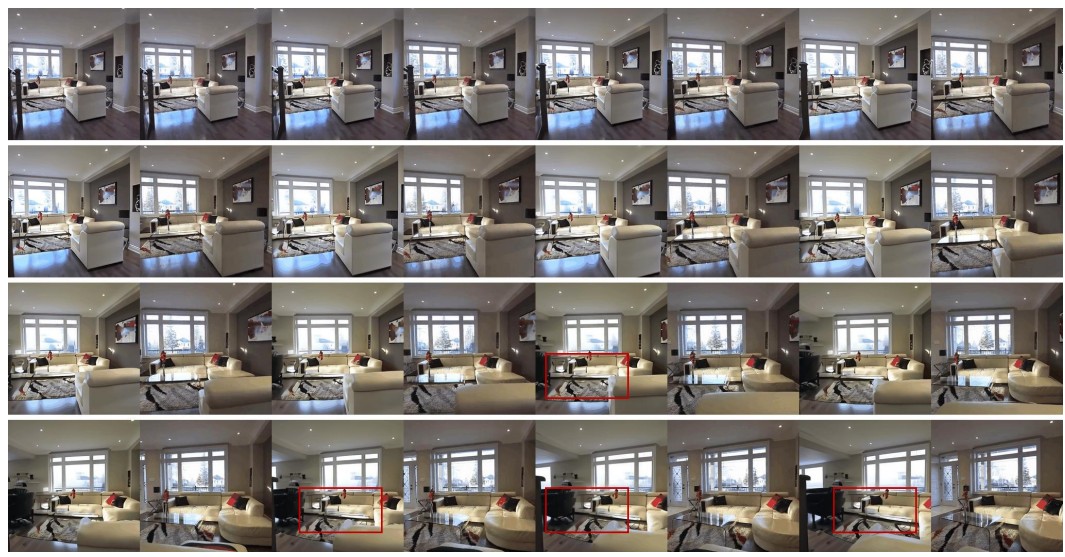

Figure 6: **Failure Case Analysis.** The transparent, reflective glass table intermittently disappears and reappears across frames, indicating that the model still has difficulty handling reflective materials. The red box indicates when the table disappears.

## F    Supplementary Visualizations

To better understand the effects of geometry, we provide comprehensive visual results.

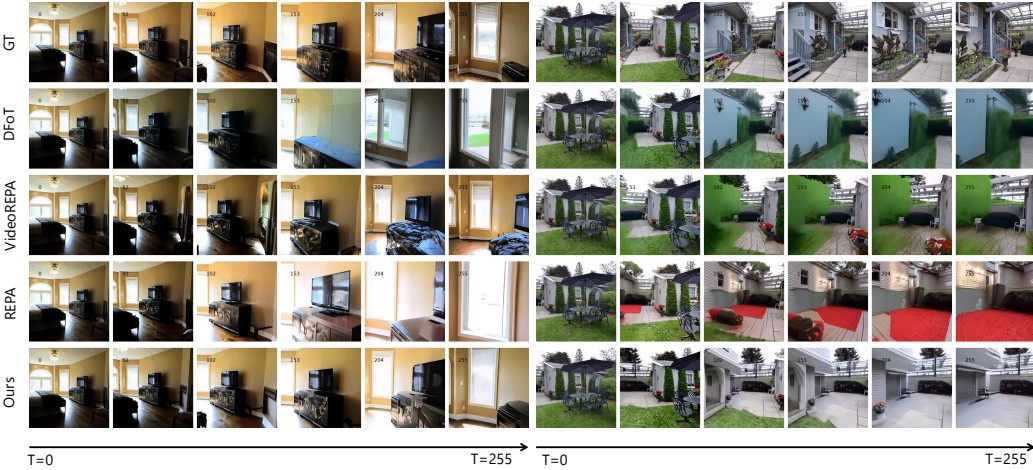

Figure 7: **Qualitative comparisons on camera-conditioned video generation.** All the videos are generated given the first frame and per-frame camera pose. We comprehensively compare GF (ours) with DFoT (Song et al., 2025), VideoREPA (Zhang et al., 2025c), and REPA (Yu et al., 2024a). The results demonstrate consistency in long-term video generation both inside (left) and outside (right) scenes.

Fig. 7 presents qualitative comparisons on the RealEstate10K dataset. Given the same first frame and per-frame camera trajectory as input, we compare our proposed GF method with three strong baselines: DFoT (Song et al., 2025), REPA (Yu et al., 2024a), and VideoREPA (Zhang et al., 2025c).

As shown in Fig. 7, our method generates visually coherent and geometrically consistent videos over long time horizons, even with limited context. In particular, GF better preserves object shapes and scene layouts that are visible in context, while generating reasonable scenes not seen in the context. In contrast, baseline models often exhibit drift, shape distortion, or abrupt transitions. These results highlight the effectiveness of internalizing geometric priors to enhance spatial and temporal consistency in video generation.

**Qualitative Ablation on Alignment loss.**    To further assess the impact of the proposed scale alignment loss, we conduct qualitative comparisons between models trained with and without this component (Fig. 8). While angular alignment alone helps maintain basic geometric coherence, the lack of scale supervision often leads to inconsistent camera motion, manifesting as unstable perspective changes or unnatural object scaling. By introducing the scale alignment loss, our method produces noticeably smoother viewpoint transitions and more reliable camera-following behavior, demonstrating its effectiveness in stabilizing multi-frame geometry.

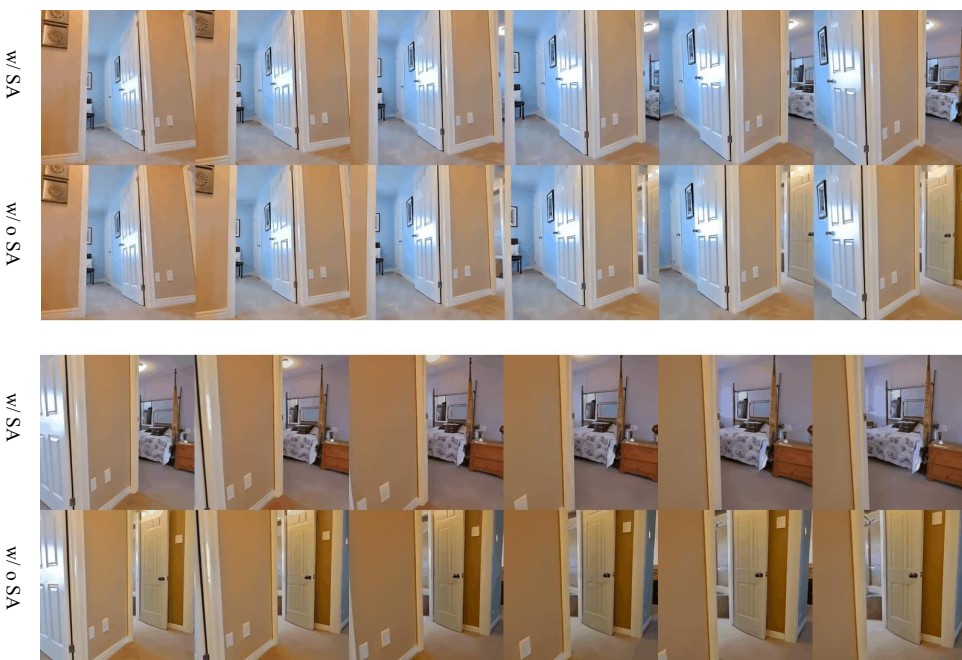

Figure 8: **Qualitative comparison of the Alignment Loss.** "w/ SA" denotes models trained with both angular alignment and scale alignment losses, while "w/o SA" refers to models trained using only angular alignment. Incorporating scale alignment enables the model to generate videos with more stable and realistic camera-following behavior.

