# OpenReview forum: "Geometry Forcing: Marrying Video Diffusion and 3D Representation for Consistent World Modeling"
_ICLR.cc/2026/Conference — ICLR 2026 Poster_

### Official Review · Reviewer_qFpu · 2025-10-15

**Soundness:** 3
**Presentation:** 3
**Contribution:** 2
**Rating:** 6
**Confidence:** 3

**Summary:**

This paper proposes Geometry Forcing, a REPA-style feature alignment method to enhance the 3D consistency of existing video diffusion models (VDMs). Authors first observe that existing VDMs cannot readout consistent 3D representations (i.e., point maps) from their diffusion features. To address this, prior works either jointly predict the 3D modality, or leverage a structured 3D representation as guidance. Geometry Forcing instead proposes to align the internal features of the VDM with the representation of a 3D foundation model, VGGT. Fine-tuning pre-trained VDMs with such auxiliary loss significantly improves the temporal consistency of the generated videos, demonstrated by both quantitative and user study results.

**Strengths:**

- The idea is simple and intuitive. Adding a REPA loss with VGGT does not introduce any sophisticated architecture design, yet it is able to bake the 3D information into the model.
- The ablation study is very comprehensive, which clearly shows the effectiveness of each module.
- The experimental results seem strong. Geometry Forcing outperforms baselines in most of the metrics.

**Weaknesses:**

1. As the author pointed out in the Limitation, VGGT is only trained on static scenes, and thus cannot be used to supervise VDM training on dynamic videos. Can authors discuss more on how to extend Geometry Forcing to dynamic videos as required by general text-to-video training?
2. Have the authors tried other 3D foundation models? For example, MonST3R [1] and CUT3R [2] as they can handle dynamic videos.
3. Have you tried training a video diffusion model from scratch? REPA shows that using representation alignment loss can greatly speed up convergence. I'm curious if applying both DINO and VGGT loss can further accelerate this. I understand that the computation cost is high, so showing some early training loss curve (with vs without Geometry Forcing loss) is enough (or describe it, if figures are not allowed in rebuttal).
4. I'd like to see a comparison with a baseline that uses explicit 3D memory, e.g., GEN3C [3] that uses reprojected point clouds as additional conditioning for the VDM. You do not need to apply all its components. I think you can do this:
- To generate frame `n`, run VGGT on frame `1, 2, ..., n-1` to reconstruct a point cloud of the scene, then render it to 2D images, and condition the VDM on it. This can also work in an autoregressive way.

[1] Zhang, Junyi, et al. "Monst3r: A simple approach for estimating geometry in the presence of motion." arXiv preprint arXiv:2410.03825 (2024).

[2] Wang, Qianqian, et al. "Continuous 3d perception model with persistent state." Proceedings of the Computer Vision and Pattern Recognition Conference. 2025.

[3] Ren, Xuanchi, et al. "Gen3c: 3d-informed world-consistent video generation with precise camera control." Proceedings of the Computer Vision and Pattern Recognition Conference. 2025.

**Questions:**

Besides the questions in Weaknesses, here are some minor questions:
1. Can you explain why VideoREPA achieves a significantly better RVE than Geometry Forcing, while being much worse in other metrics?
2. How do you implement VideoREPA? Is it doing REPA alignment loss with per-frame DINOv2 features? Then why is its result much worse than the "DINOv2 Only" entry in Tab.2?

---

> ### Author Response · Authors · 2025-11-26
> **Response 1 to Reiewer qFpu**
>
> We sincerely thank the reviewer for acknowledging our paper's idea and experiments. We also really appreciate the reviewer's valuable suggestions. We post our reply to the reviewer's questions as follows.
>
> ---
>
> > **W1:** *As the author pointed out in the Limitation, VGGT is only trained on static scenes, and thus cannot be used to supervise VDM training on dynamic videos. Can authors discuss more on how to extend Geometry Forcing to dynamic videos as required by general text-to-video training?*
>
> **R1: Dynamic Scene and Extension to Text-to-Video Training.** We thank the reviewer for raising the important question of whether VGGT, trained only on static scenes, can support Geometry Forcing in dynamic video generation. In practice, we observe that VGGT maintains stable reconstruction performance on dynamic scenes and still provides useful 3D cues for supervising VDMs.
>
> To validate this, we applied Geometry Forcing to a text-to-video setting by finetuning a pretrained Wan2.1-1.3B model. Despite motion and scene dynamics, the method consistently improved video quality.
>
> **Text-to-Video Experiment Details.** We used 10k videos from the SpatialVID dataset, which offers high-resolution videos with rich scene and camera annotations. Wan2.1-1.3B was finetuned on these videos and evaluated on a held-out set of 100 videos. Both semantic quality and aesthetic scores improved noticeably for Wan2.1-GF compared with the baseline.
>
> These results indicate that Geometry Forcing can extend effectively to dynamic text-to-video training, even though VGGT itself is trained on static scenes. Due to limited data and resources, these are just preliminary results.
>
> | Method | Aesthetic ↑ | Imaging Quality ↑ | Motion Smoothness ↑ |
> | :--- | :---: | :---: | :---: |
> | **Wan 2.1** | 0.58 | 0.56 | 0.98 |
> | **Geometry Forcing Wan 2.1** | 0.59 | 0.59 | 0.99 |
>
> > **W2:** *Have the authors tried other 3D foundation models? For example, MonST3R [1] and CUT3R [2] as they can handle dynamic videos.*
>
> **R2: Other Teacher Models.** We thank the reviewer for this valuable suggestion. When selecting a 3D foundation model for Geometry Forcing, two criteria are essential: (1) a feed-forward architecture without per-scene optimization to keep the online alignment cost manageable, and (2) the ability to process multiple input images while providing sufficiently strong 3D representations to supervise VDMs.
>
> The Dust3r model requires pair-wise image input, making the online feature extraction process time-consuming. The CUT3R model processes input images sequentially, which also becomes time-consuming as the number of input views increases.
>
> Following these principles, we experimented with Pi3 as the teacher model. Incorporating Pi3 [1] into Geometry Forcing reduced the FVD from ~360 → 309, demonstrating a substantial improvement in video generation quality.
>
> | Method | FVD ↓ | LPIPS ↓ | SSIM ↑ | PSNR ↑ | RVE ↓ | RPE ↓ |
> | :--- | :---: | :---: | :---: | :---: | :---: | :---: |
> | **Baseline** | 364 | 0.55 | 0.36 | 11.40 | 297.00 | 0.3575 |
> | **GF-VGGT** | 243 | 0.51 | 0.38 | 11.87 | 272.00 | 0.3337 |
> | **GF-Pi3** | 309 | 0.53 | 0.38 | 11.53 | 303.32 | 0.3171 |
>
> *[1] Wang, Yifan, et al. "π3: Scalable permutation-equivariant visual geometry learning." arXiv preprint arXiv:2507.13347 (2025).*
>
> > **W3:** *Have you tried training a video diffusion model from scratch? REPA shows that using representation alignment loss can greatly speed up convergence. I'm curious if applying both DINO and VGGT loss can further accelerate this. I understand that the computation cost is high, so showing some early training loss curve (with vs without Geometry Forcing loss) is enough (or describe it, if figures are not allowed in rebuttal).*
>
> **R3: Training From Scratch using GF.** We thank the reviewer for this observation. Indeed, we expect Geometry Forcing (GF) to accelerate convergence in video generation when training from scratch. However, due to limited computing resources, we were only able to obtain preliminary results at the early training stage. These initial findings demonstrate that GF achieves a much faster decrease in validation FVD compared to standard diffusion training.
>
> | Step | DFoT FVD ↓ | GF FVD ↓ |
> | :--- | :---: | :---: |
> | 15k | 2852.0 | 2396.0 |
> | 30k | 2440.0 | 939.5 |
> | 45k | 1634.0 | 356.25 |

---

> ### Author Response · Authors · 2025-11-26
> **Response 2 to Reviewer qFpu**
>
> > **W4:** *I'd like to see a comparison with a baseline that uses explicit 3D memory, e.g., GEN3C [3] that uses reprojected point clouds as additional conditioning for the VDM. You do not need to apply all its components.*
>
> **R4: Explicit Geometry-Based Video Generation Baseline.** We thank the reviewer for the suggestion. To provide a more complete comparison, we implemented explicit geometry-based baselines for video generation. Following your advice, we reconstruct the 3D scene using VGGT, render a 2D conditioning image, and inject this rendered image into the diffusion model via a ControlNet. During inference, the 3D scene is continuously updated using the initial frame and previously generated frames, and the resulting rendered images are used to condition the next stage of generation.
>
> This explicit geometric guidance improves performance over the vanilla VDM, demonstrating the benefit of incorporating geometric cues. However, these explicit baselines still underperform compared to Geometry Forcing. GF offers two key advantages:
> * **No additional inference cost:** It does not require loading or running a 3D foundation model during sampling, thereby reducing memory usage and avoiding extra computation.
> * **Higher generation quality:** Its implicit geometric supervision consistently yields better results than explicit conditioning.
>
> Overall, while explicit geometry control is helpful, Geometry Forcing remains both more effective and more efficient.
>
> | Method | FVD ↓ | LPIPS ↓ | SSIM ↑ | PSNR ↑ | RVE ↓ | RPE ↓ |
> | :--- | :---: | :---: | :---: | :---: | :---: | :---: |
> | **Baseline** | 364 | 0.55 | 0.36 | 11.40 | 297.00 | 0.3575 |
> | **Explicit Geometry**| 280 | 0.52 | 0.37 | 11.99 | 296.94 | 0.3792 |
> | **GF** | 243 | 0.51 | 0.38 | 11.87 | 272.00 | 0.3337 |
>
> > **Q1:** *Can you explain why VideoREPA achieves a significantly better RVE than Geometry Forcing, while being much worse in other metrics?*
>
> **R5: VideoREPA RVE Results.** We thank the reviewer for the detailed review! As shown in Fig. 2, VideoREPA’s rotation generation produces frames that are almost identical to the first frame. While RVE only compares the FID between input frames and revisited frames, it completely ignores intermediate frames. This allows VideoREPA to achieve a better RVE despite producing ambiguous intermediate frames and poor camera following.
>
> > **Q2:** *How do you implement VideoREPA? Is it doing REPA alignment loss with per-frame DINOv2 features? Then why is its result much worse than the "DINOv2 Only" entry in Tab.2?*
>
> **R6: Difference Between DINOv2 Only and VideoREPA.** We thank the reviewer for pointing this out. We follow the VideoREPA implementation to use VideoMAE v2 as the teacher representation model. In contrast, REPA uses a per-frame alignment loss using DINOv2 only. We believe the result gap shown in Tab. 2 mainly comes from the internal representation quality difference between VideoMAE v2 and DINOv2.

---

### Official Review · Reviewer_J25o · 2025-10-20

**Soundness:** 3
**Presentation:** 3
**Contribution:** 2
**Rating:** 4
**Confidence:** 3

**Summary:**

The paper introduces Geometry Forcing (GF) that enhances the geometric consistency of video diffusion models by aligning their internal representations with those of a 3D foundation model (VGGT). GF introduces Angular Alignment and Scale Alignment. The method is integrated into standard autoregressive video diffusion training without requiring explicit 3D supervision. Experiments on RealEstate10K and Minecraft benchmarks show that GF improves Fréchet Video Distance (FVD), SSIM, and 3D consistency metrics (RPE/RVE) compared to state-of-the-art baselines such as DFoT, REPA, and VideoREPA. Ablation studies confirm the complementary role of Angular and Scale alignment, and user studies indicate perceptually improved scene consistency.

**Strengths:**

1. The paper is very well written, with clear figures illustrating motivation and results.

2. The paper proposes a new concept of “geometry forcing”, transferring geometric awareness into video diffusion models without requiring 3D ground-truth supervision. The dual-objective design (Angular and Scale alignment) is interesting, addressing optimization stability in cross-domain feature matching.

3. Experiments are extensive. Quantitative: Benchmarks include both 16- and 256-frame video generation, using perceptual and geometric metrics. Qualitative: Visual comparisons (360° rotations) convincingly show consistent viewpoint revisiting.

**Weaknesses:**

1. Scale of experiments is modest (16–256 frames, 256×256 resolution). The authors acknowledge this but it limits claims of scalability.

2. While the ablations cover alignment types and layer depths, computational cost (training overhead, memory footprint) is missing. Geometry alignment likely adds feature extraction and projection costs that may be nontrivial.

3. The method relies heavily on VGGT as a teacher. It’s unclear whether GF’s success depends on the specific 3D foundation model or generalizes to others (e.g., DUST3R, FLARE).

4. There is limited discussion of failure cases (e.g., ambiguous depth, reflective surfaces).

**Questions:**

1. Generality of the 3D teacher. How sensitive is GF to the choice of 3D foundation model? Would a weaker teacher (e.g., DUST3R) still yield benefits, or is VGGT’s strong geometry essential?

2. Does alignment at all layers or at multiple scales (spatial or temporal) offer further improvement, beyond the mid-level layer shown to be best?

3. What is the added training cost (e.g., % increase in FLOPs or wall-clock time) from Geometry Forcing, given the need to compute VGGT features?

---

> ### Author Response · Authors · 2025-11-26
> **Response to Reviewer J25o**
>
> # 03 Response to Reviewer J25o
>
> We sincerely thank the reviewer for acknowledging our paper's writing, technical novelty, and experiments. We also really appreciate the reviewer's valuable suggestions. We post our reply to the reviewer's questions as follows.
>
> ---
>
> > **W1:** *Scale of experiments is modest (16–256 frames, 256×256 resolution). The authors acknowledge this but it limits claims of scalability.*
>
> **R1: Scale of Experiments.** We thank the reviewer for the comment. Due to limited computational resources, we were unable to train on larger resolutions or very long sequences. However, our method introduces only a lightweight alignment loss and does not modify the model architecture, so it is directly compatible with higher-resolution and longer-duration training. We expect the approach to scale naturally with stronger backbones and larger training budgets.
>
> > **W2:** *While the ablations cover alignment types and layer depths, computational cost (training overhead, memory footprint) is missing. Geometry alignment likely adds feature extraction and projection costs that may be nontrivial.*
>
> > **Q3:** *What is the added training cost (e.g., % increase in FLOPs or wall-clock time) from Geometry Forcing, given the need to compute VGGT features?*
>
> **R2: Feature Extraction and Projection Costs.** We appreciate the reviewer for raising this important point regarding computational costs. Our profiling results show that feature extraction for 16 frames requires approximately 0.8 seconds on a single A800 GPU, with the processing time scaling linearly with the number of frames. During Geometry Forcing training, the forward pass through VGGT takes 0.853 seconds, the projection operation requires 0.017 seconds, and the backward pass accounts for 0.5 seconds.
>
> We would like to emphasize that these feature extraction costs are exclusively incurred during training and do not affect inference efficiency. This represents a key advantage of our approach compared to explicit conditioning methods, as our method introduces no additional computational burden to the generation process at inference time.
>
> Detailed wall-clock time and FLOPs profiling results are provided in Appendix E.1.
>
> > **W3:** *The method relies heavily on VGGT as a teacher. It’s unclear whether GF’s success depends on the specific 3D foundation model or generalizes to others (e.g., DUST3R, FLARE).*
>
> > **Q1:** *Generality of the 3D teacher. How sensitive is GF to the choice of 3D foundation model? Would a weaker teacher (e.g., DUST3R) still yield benefits, or is VGGT’s strong geometry essential?*
>
> **R3: Generalizability to Other Teacher Models.** We thank the reviewer for the suggestion. The generality of Geometry Forcing across 3D foundation models is important, but not all models are suitable for online alignment. For example, DUST3R requires per-scene optimization and cannot provide feed-forward features, making it impractical. In contrast, end-to-end feed-forward models like Pi3 are efficient and compatible. Specifically, the DUST3R model requires pair-wise image input, which makes the online feature extraction process time-consuming. The CUT3R model processes input images sequentially, which also becomes time-consuming as the number of input views increases.
>
> We conducted an experiment using Pi3 as the teacher. Aligning the diffusion model with Pi3 reduced FVD from 360 → 309, showing significant improvement (see Appendix D.1, Table 7). This demonstrates that Geometry Forcing generalizes to other suitable 3D foundation models while keeping inference costs manageable.
>
> | Method | FVD ↓ | LPIPS ↓ | SSIM ↑ | PSNR ↑ | RVE ↓ | RPE ↓ |
> | :--- | :---: | :---: | :---: | :---: | :---: | :---: |
> | **Baseline** | 364 | 0.55 | 0.36 | 11.40 | 297.00 | 0.3575 |
> | **GF-VGGT** | 243 | 0.51 | 0.38 | 11.87 | 272.00 | 0.3337 |
> | **GF-Pi3** | 309 | 0.53 | 0.38 | 11.53 | 303.32 | 0.3171 |
>
> > **W4:** *There is limited discussion of failure cases (e.g., ambiguous depth, reflective surfaces).*
>
> **R4: Failure Case Analysis.** We thank the reviewer for highlighting the challenge posed by reflective objects. While our method substantially improves visual quality and geometric consistency, certain complex scenarios remain difficult. As shown in the updated Fig. 6 of the paper, the transparent and reflective glass table intermittently disappears and reappears across frames, revealing that the model still struggles with handling highly reflective materials. We appreciate the reviewer’s attention to this limitation and have emphasized this failure case more clearly in the revision.

---

> > ### Author Response · Authors · 2025-11-27
> > **Response2 to Reviewer J25o**
> >
> > > **Q2:** *Does alignment at all layers or at multiple scales (spatial or temporal) offer further improvement, beyond the mid-level layer shown to be best?*
> >
> > **R5: Number of Alignment Layers.** We thank the reviewer for raising this point. We experimented with aligning different layers of the diffusion model, such as the last three layers. Interestingly, while aligning a single layer already provides improvement, aligning multiple layers simultaneously does not always yield better results. Due to the large number of possible layer combinations, we report results for aligning the last three layers in Table 10. Full details are provided in Appendix D.4.
> >
> > | Method | FVD ↓ | LPIPS ↓ | SSIM ↑ | PSNR ↑ | RVE ↓ | RPE ↓ |
> > | :--- | :---: | :---: | :---: | :---: | :---: | :---: |
> > | **Baseline** | 364 | 0.55 | 0.36 | 11.40 | 297.00 | 0.3575 |
> > | **GF-Multiple-Layer** | 271 | 0.51 | 0.37 | 12.03 | 295.07 | 0.3133 |
> > | **GF** | 243 | 0.51 | 0.38 | 11.87 | 272.00 | 0.3337 |

---

### Official Review · Reviewer_PFvJ · 2025-10-31

**Soundness:** 2
**Presentation:** 2
**Contribution:** 2
**Rating:** 4
**Confidence:** 4

**Summary:**

This paper adapts the REPresentation Alignment (REPA) [1] into 3D domain. Specifically, it employs VGGT [2] as a 3D foundation model and aligns the intermediate features of a video diffusion model with features extracted from layers of VGGT. The authors propose two alignment objectives for this purpose: an angular alignment objective, which enforces cosine similarity between feature maps, and a scale alignment objective, which directly supervises the rescaled features of the diffusion model. Experiments demonstrates that combining these two objectives enhances the video diffusion model's 3D understanding, thereby improving geometric consistency in the generated results.

[1] Representation Alignment for Generation: Training Diffusion Transformers Is Easier Than You Think

[2] VGGT: Visual Geometry Grounded Transformer

**Strengths:**

* The core concept of aligning intermediate features of a generative model with those of a foundation model is a promising research direction that has shown success in other fields.
* The proposed method is simple yet effective. The experiments demonstrate improvements over the baseline models.
* The paper is well-written and easy to understand.

**Weaknesses:**

* My primary concern is the paper's limited novelty. The proposed method appears to be a straightforward adaptation of the 2D alignment technique from REPA. The core contribution seems to be replacing the DINO model (used in 2D REPA) with VGGT for the 3D case. As such, the work feels incremental and offers limited new conceptual insights.

* The method's applicability appears to be limited to static scenes, but this is not explicitly stated. The authors should clearly acknowledge that the current approach does not handle dynamic scenes or significant camera motion, which restricts its use to a narrow set of conditions.

*  The performance of the VGGT feature extraction likely depends on the number of views used. I would encourage the authors to provide an ablation study demonstrating how model performance varies with different numbers of views adopted during training time.

**Questions:**

*  The matching process of VGGT seems computationally expensive, especially as the number of views increases, potentially making the training alignment prohibitively slow. To clarify this, could the authors report the concrete training time?

---

> ### Author Response · Authors · 2025-11-26
> **Response to Reviewer PFvj**
>
> We sincerely thank the reviewer for acknowledging our paper's writing, concept, and experiments. We also really appreciate the reviewer's valuable suggestions. We post our reply to the reviewer's questions as follows.
>
> ---
>
> > **W1:** *My primary concern is the paper's limited novelty. The proposed method appears to be a straightforward adaptation of the 2D alignment technique from REPA. The core contribution seems to be replacing the DINO model (used in 2D REPA) with VGGT for the 3D case. As such, the work feels incremental and offers limited new conceptual insights.*
>
> **R1: Paper Novelty.** We thank the reviewer for the comment. Several reviewers (AqJj, J25o, qFpu) noted that the method is simple and effective, with Reviewer J25o highlighting that our dual-objective design (Angular + Scale alignment) offers a meaningful improvement in cross-domain feature matching.
>
> Our contribution goes beyond a direct extension of 2D REPA. To the best of our knowledge, we are the first to show that aligning a video diffusion model with a latent 3D foundation model can substantially enhance both video quality and 3D consistency. This provides a new perspective: 3D representation learning directly benefits video generation.
>
> Technically, our novelty lies in:
> - Proposing Geometry Forcing, which internalizes 3D structure without extra inference cost; and
> - Introducing a dual-objective (Scale + Angular) alignment that stabilizes and strengthens cross-domain feature alignment.
>
> Thus, while the idea is intentionally simple, it offers new insight and practical impact in bringing 3D understanding into video diffusion models.
>
> > **W2:** *The method's applicability appears to be limited to static scenes, but this is not explicitly stated. The authors should clearly acknowledge that the current approach does not handle dynamic scenes or significant camera motion, which restricts its use to a narrow set of conditions.*
>
> **R2: Dynamic Scene and Large Camera Motion.** Thank you for the insightful comment. Our method is not limited to static scenes: GF aligns geometric representations at the feature level and does not impose assumptions about scene rigidity. In practice, we observe that the method handles both object motion and large camera motion well. As shown in Fig. 2, the model can generate 360° rotations from a single frame, demonstrating robustness to significant viewpoint changes.
>
> To further validate this, we conducted additional experiments on a dynamic video dataset, where GF continues to improve video quality over the baseline. These results have been added to Appendix D.5. Due to limited data and resources, these are just preliminary results.
>
> | Method | Aesthetic ↑ | Imaging Quality ↑ | Motion Smoothness ↑ |
> | :--- | :---: | :---: | :---: |
> | **Wan 2.1** | 0.58 | 0.56 | 0.98 |
> | **Geometry Forcing Wan 2.1**| 0.59 | 0.59 | 0.99 |
>
> > **W3:** *The performance of the VGGT feature extraction likely depends on the number of views used. I would encourage the authors to provide an ablation study demonstrating how model performance varies with different numbers of views adopted during training time.*
>
> **R3: Ablation on Number of Views.** Thank you for the helpful suggestion. We conducted an ablation study using 4, 8, and 16 alignment views, and the results are included in Appendix D.3 (Table 9). The per-step training time changes only slightly across different view counts, confirming that multi-view alignment does not introduce significant overhead. However, using more views provides more complete 3D scene information and leads to consistently better video generation quality.
>
> | Method | FVD ↓ | LPIPS ↓ | SSIM ↑ | PSNR ↑ | RVE ↓ | RPE ↓ |
> | :--- | :---: | :---: | :---: | :---: | :---: | :---: |
> | **Baseline** | 364 | 0.55 | 0.36 | 11.40 | 297 | 0.3575 |
> | **Context = 4** | 261 | 0.51 | 0.38 | 12.21 | 297 | 0.3451 |
> | **Context = 8** | 257 | 0.50 | 0.38 | 12.17 | 284 | 0.3062 |
> | **Context = 16 (GF)** | 243 | 0.51 | 0.38 | 11.87 | 272 | 0.3337 |

---

> > ### Author Response · Authors · 2025-11-27
> > **Response2 to Reviewer PFvj**
> >
> > > **Q1:** *The matching process of VGGT seems computationally expensive, especially as the number of views increases, potentially making the training alignment prohibitively slow. To clarify this, could the authors report the concrete training time?*
> >
> > **R4: Training Cost Analysis.** We appreciate the reviewer’s concern. In practice, the alignment process is not computationally expensive. VGGT is a feed-forward model that processes all reference views in a single forward pass, and we use at most 16 reference images per video, keeping the alignment overhead manageable.
> >
> > On our setup, each training step of the base diffusion model takes approximately 1.0 second, and adding VGGT alignment increases this to about 2.0 seconds. The full training of 20k iterations finishes within a few hours on 8×A100 GPUs. The feature extraction time from 1 to 16 views is 0.1 second to 0.8 seconds.
> >
> > We perform a detailed profiling of our method on an NVIDIA A800 GPU and report both the execution time and floating-point operations (FLOPs) for different components of our model during the training stage in Table 11. The VGGT Feature Alignment contributes an additional 52.5% in execution time and 60.4% in total FLOPs. Although this alignment process increases the per-step computation compared to the base diffusion model, it significantly accelerates convergence, thereby reducing the overall training duration. For fine-tuning, our method requires only a few thousand steps and completes within hours. Additionally, during inference, our method does not introduce any additional computational cost compared to other methods that use explicit or implicit memory.
> >
> > Overall, the alignment overhead is moderate and does not pose a practical limitation.

---

### Official Review · Reviewer_AqJj · 2025-11-01

**Soundness:** 3
**Presentation:** 3
**Contribution:** 3
**Rating:** 4
**Confidence:** 4

**Summary:**

This work introduces Geometry Forcing, a method designed to help video diffusion models better capture the inherent 3D structure of real-world scenes. While standard video diffusion models trained on raw 2D video data often lack geometric understanding, Geometry Forcing addresses this by aligning the model’s internal representations with features from a geometric foundation model. It employs two key alignment strategies: Angular Alignment, which enforces directional consistency through cosine similarity, and Scale Alignment, which preserves scale information via feature regression. Experiments on camera view–conditioned and action-conditioned video generation show that Geometry Forcing significantly enhances both visual quality and 3D consistency, outperforming existing baselines.

**Strengths:**

1. The paper is clearly written and well-organized.
2. The core idea is simple yet effective — aligning the internal representations of video diffusion models with features from a geometric foundation model.
3. Experimental results demonstrate strong performance, showing notable improvements in geometric consistency and long-term temporal coherence compared to baseline methods.

**Weaknesses:**

The training objectives of the diffusion model and the VGGT differ fundamentally. The diffusion model is designed to learn noise or velocity in a progressive manner—its target lies in the intermediate denoising process rather than the final outcome. In contrast, VGGT is result-oriented, directly learning to predict the final geometry. Although the experimental results appear promising, theoretically the learning targets are not of the same nature and may even be somewhat conflicting. It remains unclear how the proposed alignment between these two objectives effectively works in practice. Could the authors provide more theoretical or empirical justification for this compatibility?
The motivations and formulations of Angular Alignment and Scale Alignment are insufficiently explained. The directional correspondence between the hidden states of the diffusion model and the geometric features, as well as the scale differences across models, are both vague. It would be helpful to clarify how these factors influence the final generation quality.

The base model description is also unclear. The paper mentions “a U-ViT backbone for video generation,” but does not specify which model this refers to or provide an appropriate citation.

Including explicit geometry-based video generation methods as comparison baselines would strengthen the evaluation and provide more persuasive evidence of the proposed method’s effectiveness.

Additionally, there are citation errors, such as the one noted around line 290.

Finally, the paper lacks qualitative ablation studies that isolate the effects of Angular Alignment and Scale Alignment. Without such analyses, it is difficult to assess the actual contributions of each component to the overall improvement.

**Questions:**

1.The diffusion model and VGGT have different learning objectives — one is progressive (learning noise or velocity), while the other is result-oriented. How can the proposed alignment between these fundamentally different targets work effectively?


2.The motivations and mechanisms of Angular Alignment and Scale Alignment are unclear. How exactly do these objectives influence the final video generation quality?


3.What specific model does the “U-ViT backbone for video generation” refer to? Could the authors provide more details or a proper citation?

4. Could the authors include explicit geometry-based video generation methods as additional baselines to make the comparison more convincing?


5.The paper lacks qualitative ablation studies showing the individual effects of Angular and Scale Alignment. Can the authors provide such qualitative analyses?

---

> ### Author Response · Authors · 2025-11-26
> **Response to Reviewer AqJj**
>
> We sincerely thank the reviewer for acknowledging our paper's writing, idea, and experiments, and we appreciate the reviewer's valuable suggestions.
>
> ---
>
> > **W1 (part 1):** *The training objectives of the diffusion model and the VGGT differ fundamentally. The diffusion model is designed to learn noise or velocity in a progressive manner—its target lies in the intermediate denoising process rather than the final outcome. In contrast, VGGT is result-oriented, directly learning to predict the final geometry. Although the experimental results appear promising, theoretically the learning targets are not of the same nature and may even be somewhat conflicting. It remains unclear how the proposed alignment between these two objectives effectively works in practice. Could the authors provide more theoretical or empirical justification for this compatibility?*
>
> > **Q1:** *The diffusion model and VGGT have different learning objectives — one is progressive (learning noise or velocity), while the other is result-oriented. How can the proposed alignment between these fundamentally different targets work effectively?*
>
> **R1: Different Learning Objectives.** We thank the reviewer for raising this important question. To the best of our knowledge, we are the first to demonstrate that aligning a generation model with a 3D-aware feature space using a dual-objective design is effective, which is non-trivial.
>
> While the learning objectives differ, their targets are complementary: diffusion models learn the implicit distribution of image or video data, while 3D reconstruction models like VGGT learn to reconstruct 3D geometry from 2D images. Our Geometry Forcing can be viewed as an alignment process that adds 3D information to help the diffusion model learn to generate videos. Specifically, the 3D features from VGGT serve as a strong geometric prior to guide the diffusion process.
>
> For instance, REPA [1] successfully aligned a diffusion model to the semantic feature space of DINOv2 to significantly improve generation quality and semantic control. Similarly, in our case, we demonstrate that aligning with a 3D-aware feature space from VGGT provides an effective 3D structural prior for video generation. The key is the transfer of robust feature understanding, not the specific form of the source model's loss function.
>
> *[1] Yu, Sihyun, et al. "Representation alignment for generation: Training diffusion transformers is easier than you think." arXiv preprint arXiv:2410.06940 (2024).*
>
> > **W1 (part 2):** *The motivations and formulations of Angular Alignment and Scale Alignment are insufficiently explained. The directional correspondence between the hidden states of the diffusion model and the geometric features, as well as the scale differences across models, are both vague. It would be helpful to clarify how these factors influence the final generation quality.*
>
> > **Q2:** *The motivations and mechanisms of Angular Alignment and Scale Alignment are unclear. How exactly do these objectives influence the final video generation quality?*
>
> > **W5:** *Finally, the paper lacks qualitative ablation studies that isolate the effects of Angular Alignment and Scale Alignment. Without such analyses, it is difficult to assess the actual contributions of each component to the overall improvement.*
>
> > **Q5:** *The paper lacks qualitative ablation studies showing the individual effects of Angular and Scale Alignment. Can the authors provide such qualitative analyses?*
>
> **R2: Angular and Scale Alignment.** We thank the reviewer for the insightful suggestion.
>
> As discussed in Sec. 4.2, our alignment objectives are introduced to bridge the gap between the video latent space and the 3D feature space, a mismatch that often leads to temporally unstable or geometrically inconsistent video generation.
>
> **Motivation & Mechanism.** Angular Alignment is designed to enforce directional consistency between latent features and their corresponding 3D representations. However, angular consistency alone is insufficient for resolving ambiguities related to magnitude or depth scaling. To address this, Scale Alignment is introduced as a complementary objective that aligns scale-related information, enabling the model to infer more accurate camera motion and depth progression across frames.
>
> **Qualitative Ablation of Angular and Scale Alignment.** We provide the quantitative ablation of Angular and Scale Alignment in Table 3. As shown, Angular Alignment provides a strong baseline improvement, while adding Scale Alignment consistently yields further gains across all metrics.

---

> > ### Author Response · Authors · 2025-11-27
> > **Response2 to Reviewer AqJj**
> >
> > **Quantitative Results and Impact on Video Quality.** Beyond numbers, we now include a new qualitative comparison (updated in the main paper as Fig. 8, and expanded in Appendix F). These visual results clearly demonstrate that models trained with only angular alignment occasionally fail to follow the intended camera trajectory, producing unstable perspective shifts. By contrast, incorporating the scale alignment loss results in videos with noticeably more stable geometry and more faithful camera-following behavior. Due to limited data and resources, these are just preliminary results.
> >
> > > **W2:** *The base model description is also unclear. The paper mentions “a U-ViT backbone for video generation,” but does not specify which model this refers to or provide an appropriate citation.*
> >
> > > **Q3:** *What specific model does the “U-ViT backbone for video generation” refer to? Could the authors provide more details or a proper citation?*
> >
> > **R3: U-ViT Backbone.** Thank you for pointing this out. This citation was present in our original manuscript at Line 423.
> >
> > > **W3:** *Including explicit geometry-based video generation methods as comparison baselines would strengthen the evaluation and provide more persuasive evidence of the proposed method’s effectiveness.*
> >
> > > **Q4:** *Could the authors include explicit geometry-based video generation methods as additional baselines to make the comparison more convincing?*
> >
> > **R4: Explicit Geometry-Based Video Generation Baseline.** We thank the reviewer for the suggestion. To provide a more complete comparison, we implemented explicit geometry-based baselines for video generation. Following Reviewer qFpu’s advice, we reconstruct the 3D scene using VGGT, render a 2D conditioning image, and inject this rendered image into the diffusion model via a ControlNet. During inference, the 3D scene is continuously updated using the initial frame and generated frames, and the rendered images are used to condition the next stage of generation.
> >
> > This explicit geometric guidance improves performance over the vanilla VDM, demonstrating the benefit of incorporating geometric cues. However, these explicit baselines still underperform compared to Geometry Forcing. GF offers two key advantages:
> > * **No additional inference cost:** It does not require loading or running a 3D foundation model during sampling, thereby reducing memory usage and avoiding extra computation.
> > * **Higher generation quality:** Its implicit geometric supervision consistently yields better results than explicit conditioning.
> >
> > Overall, while explicit geometry control is helpful, Geometry Forcing remains both more effective and more efficient.
> >
> > | Method | FVD ↓ | LPIPS ↓ | SSIM ↑ | PSNR ↑ | RVE ↓ | RPE ↓ |
> > | :--- | :---: | :---: | :---: | :---: | :---: | :---: |
> > | **Baseline** | 364 | 0.55 | 0.36 | 11.40 | 297.00 | 0.3575 |
> > | **Explicit Geometry**| 280 | 0.52 | 0.37 | 11.99 | 296.94 | 0.3792 |
> > | **GF-VGGT** | 243 | 0.51 | 0.38 | 11.87 | 272.00 | 0.3337 |
> >
> > > **W4:** *Additionally, there are citation errors, such as the one noted around line 290.*
> >
> > **R5: Citation Errors.** We apologize for the oversight. We have corrected the citation error in the revised manuscript.

---

### Meta-Review · Area_Chair_wrc5 · 2026-01-06

**Summary:**

The reviewers were mostly leaning towards borderline rejection, though the rebuttal has addressed several key weaknesses (baselines / ablations / generalizability / cost analysis). While the method is straightforward, it will be a useful addition to the community, leading me to accept the paper.

**Reviewer Concerns:**

See above & below.

**Reviewer Scores:**

AqJj: 4 --> positive trend. The authors added the explicit geometry baseline and qualitative ablations.
PFvJ: 4 --> slightly positive / borderline. Concerns regarding dynamic scenes and view dependency were addressed with new experiments.
J25o: 4 --> 6 main points have been addressed.
qFpu: 6 (already accept & authors addressed baselines & generalization to Pi3 in rebuttal.

---

### Decision · Program_Chairs · 2026-01-26

Accept (Poster)